# Transcriptomic Analyses of Brains of RBM8A Conditional Knockout Mice at Different Developmental Stages Reveal Conserved Signaling Pathways Contributing to Neurodevelopmental Diseases

**DOI:** 10.3390/ijms24054600

**Published:** 2023-02-27

**Authors:** Colleen McSweeney, Miranda Chen, Fengping Dong, Aswathy Sebastian, Derrick James Reynolds, Jennifer Mott, Zifei Pei, Jizhong Zou, Yongsheng Shi, Yingwei Mao

**Affiliations:** 1Department of Biology, Pennsylvania State University, University Park, PA 16802, USA; 2Department of Genetics, Washington University School of Medicine, St. Louis, MO 63110, USA; 3Department of Biochemistry and Molecular Biology, Pennsylvania State University, University Park, PA 16802, USA; 4Department of Microbiology & Molecular Genetics, School of Medicine, University of California, Irvine, CA 92697, USA; 5Systems Biology Center, National Heart, Lung, and Blood Institute, NIH, Bethesda, Rockville, MD 20892, USA

**Keywords:** exon junction complex, nonsense-mediated decay, RBM8A, RNAseq, autism, schizophrenia

## Abstract

RNA-binding motif 8A (RBM8A) is a core component of the exon junction complex (EJC) that binds pre-mRNAs and regulates their splicing, transport, translation, and nonsense-mediated decay (NMD). Dysfunction in the core proteins has been linked to several detriments in brain development and neuropsychiatric diseases. To understand the functional role of *Rbm8a* in brain development, we have generated brain-specific *Rbm8a* knockout mice and used next-generation RNA-sequencing to identify differentially expressed genes (DEGs) in mice with heterozygous, conditional knockout (cKO) of *Rbm8a* in the brain at postnatal day 17 (P17) and at embryonic day 12. Additionally, we analyzed enriched gene clusters and signaling pathways within the DEGs. At the P17 time point, between the control and cKO mice, about 251 significant DEGs were identified. At E12, only 25 DEGs were identified in the hindbrain samples. Bioinformatics analyses have revealed many signaling pathways related to the central nervous system (CNS). When E12 and P17 results were compared, three DEGs, *Spp1*, *Gpnmb*, and *Top2a*, appeared to peak at different developmental time points in the *Rbm8a* cKO mice. Enrichment analyses suggested altered activity in pathways affecting cellular proliferation, differentiation, and survival. The results support the hypothesis that loss of *Rbm8a* causes decreased cellular proliferation, increased apoptosis, and early differentiation of neuronal subtypes, which may lead ultimately to an altered neuronal subtype composition in the brain.

## 1. Introduction

The maturation of RNA transcripts is a tightly regulated process. Pre-mRNAs usually undergo extensive modifications including splicing, polyadenylation at the 3′ end, and addition of guanosine nucleotide cap at the 5′ end before becoming translatable, mature mRNA. Diverse groups of RNA-binding proteins (RNPs) are responsible for these different RNA modifications and control RNA splicing, transport, translation, and stability, within the cell.

RBM8A, also known as Y14, is a protein that was first identified by its RNA-binding sequence [1]. RBM8A participates in an assembly of proteins known as the Exon Junction Complex (EJC), which contains the protein factors eukaryotic translation initiation factor 4A3 (EIF4A3), Magoh, cancer susceptibility candidate 3 (Casc3), and many other peripherally associated factors [2]. The EJC and its general functions are conserved across a wide range of species, with homologs being studied in different models including yeast, fly, worm, zebrafish, mouse, and human [3,4,5,6,7,8,9,10,11]. Spliceosomes assemble the EJC on spliced pre-mRNA [12]. The EJC can direct further splicing and regulate transcription or mRNA transport and translation when it accompanies the mature transcript out of the nucleus [2]. In addition to binding and modifying transcripts, the EJC has been shown to participate in Nonsense Mediated mRNA Decay (NMD), which identifies mRNA with premature termination codons (PTCs) during translation and causes the faulty mRNA to be degraded. The core components of the EJC also play their independent roles and bind to differential targets out of the EJC complex [3,13,14].

*RBM8A* mutations have been implicated in a variety of clinical phenotypes. Compound mutations in *RBM8A* have been found to cause thrombocytopenia with absent radius syndrome (TAR syndrome) [15,16,17]. This disorder is characterized primarily by low blood platelet counts (thrombocytopenia), and missing radii bones. Additional features of TAR patients include short ulnas, low megakaryocyte numbers, the axial root of the kidney, renal and heart defects, agenesis of the corpus callosum, and hypoplasia of the cerebellum [18,19,20,21]. In a case study, a TAR patient exhibited partial seizures, psychomotor retardation, and cerebral dysgenesis [20]. The genetic cause of TAR syndrome was found to have compound mutations with a microdeletion of around 200 bp in the 1q21.1 region of the genome (including *RBM8A*) on one inherited chromosome, and a low-frequency noncoding SNP in *RBM8A* on the other inherited chromosome 1 (*rs139428292* or *rs201779890*) [15,17].

In addition to clinical phenotypes of TAR syndrome, *RBM8A* is also associated with various neuropsychiatric disease cases. *RBM8A* is located in the 1q21.1 region of the genome, which is highly associated with neuropsychiatric diseases as a result of copy number variations (CNVs) (both duplication and deletions) [22,23,24,25]. Additionally, de novo mutations in *RBM8A* have been associated with autism spectrum disorders (ASD) [26] and the Mayer–Rokitansky–Küster–Hauser (MRKH) syndrome (MIM 277000) [27,28]. However, how different variants of *RBM8A* give rise to different clinical symptoms remains unknown.

To investigate the role of RBM8A in the nervous system, our lab previously demonstrated that the mouse homolog *Rbm8a* is crucial in regulating neural progenitor cell (NPC) populations and that genes downstream of *Rbm8a* expression include risk genes for intellectual disability, schizophrenia, and autism spectrum disorder [29]. Dysregulation of RBM8A leads to anxiety behaviors [30]. Consistent with its essential role in neurodevelopment, we and other groups have developed *Rbm8a* cKO mouse lines and showed that *Rbm8a* is required for the proliferation of cortical NPC and interneuron progenitors at the ganglionic eminence as well as megakaryocyte differentiation [31,32,33]. However, the underlying molecular mechanism causing these defects is still unclear. The p53 activation has been shown to mediate the cell cycle defects observed in the EJC cKO mice [33,34,35].

To further examine how the downstream molecular mechanism of *Rbm8a* causes abnormal development of the brain at different developmental periods, in this study, we analyze the changes in the transcriptome of mice with *Rbm8a* haploinsufficiency in the brain during embryonic and postnatal stages. We identified over 300 transcripts that showed significant fold changes between WT and *Rbm8a* cKO mice, including 34 genes with known functions in nervous system development. This provides a starting point for choosing a narrower subset of genes or cellular processes to observe in future studies. We further observed that neural transcription factors were upregulated in the early postnatal brain, accompanied by gene expression typically associated with mature neurons in the adult brain. Considering these results, we believe that *Rbm8a* is required to delay cell differentiation and maturation, allowing the precursor cells of the nervous system to proliferate and fully populate their organs.

## 2. Results

### 2.1. Rbm8a cKO Mouse Model

Our previous results indicate that RBM8A is essential for neural development, and more specifically, is a positive regulator of NPC proliferation [29]. However, these observed effects are limited to a small portion of the cortex, due to the limitations of in utero electroporation. To further probe this developmental role of RBM8A, and to examine its effects on the entirety of the nervous system, we generated a cKO mouse [31]. The mouse line contains the homozygous loxP allele, *Rbm8a^f/f^*, on a C57BL/6 background (Figure 1A). The *Rbm8a*^f/f^ mice contain loxP sites that guide Cre recombinase to delete three exons in the Rbm8a gene (Figure 1A). To create brain-specific Rbm8a cKO mice, the Rbm8a^f/f^ mice were crossed with *nestin-Cre (Nes-Cre*) transgenic mice from the Jackson Laboratory, *B6.Cg-Tg (Nes-Cre*) *1 Kln/J,* stock number 003771 [36]. The *Nes-Cre* mouse line has hemizygous Cre recombinase driven by a nestin promoter. Nestin has heavily biased expression in embryonic neural stem cells, allowing nervous system-specific expression of Cre at early embryonic day 10 (E10). This enabled us to examine all of the cortex, and other areas of the nervous system, and to examine how *Rbm8a* deletion in the brain affected mouse brain structure and behavior. Although nestin has been reported in a few cells in the heart or kidney, our study used the brain tissues for RNAseq to avoid contamination of other cells.

The resulting progeny consisted of 50% *Nes-Cre; Rbm8a^f/+^* mice and 50% *Rbm8a^f/+^* mice. This indicates that the mice that are haploinsufficient for *Rbm8a* are born at the expected Mendelian ratio. Littermates without nestin can be used as comparative controls. As reported previously [31], the resulting *Rbm8a* haploinsufficient mice were significantly smaller compared to littermate controls (Figure 1B) and had microcephaly, which is a greater than 50% reduction in brain size at P17 (Figure 1C). A large, visible gap between the two cerebral hemispheres was typical of the cKO brains, in contrast to the tightly aligned hemispheres in the WT brains. Most of these *Nes-cre; Rbm8a^f/+^* pups only survived until postnatal day 20 (P20). As these mice have thin cortices, we hypothesized that they also had perturbations in the cortical layers. This could manifest in the form of thinner layers, or disorganized cortical layers (cells migrating to the wrong layer). To test this, we immunostained the coronal brain section of P17 *Nes-Cre; Rbm8a^f/+^* mice and littermate controls with deep cortical layer marker FOXP2. FOXP2 staining was revealed to be abnormal; instead of staining layers 5/6 as in the control, FOXP2 labeling was found in the middle cortex, spanning to layers 3–6 (Figure 1D).

### 2.2. General DEG Analysis of the Whole Brain at P17

Next, we sought to determine the molecular pathways that govern *Rbm8a’s* role in brain development. To do this, we utilized RNAseq to determine transcriptomic changes in *Rbm8a* haploinsufficient mouse brains at P17. RNA was isolated from the whole brain of P17 mice (control and cKO) and converted to cDNA and sequences using the Illumina HiSeq 2500. In the P17 whole brain, 19,622 genes have quantifiable transcript readings that were plotted in a volcano plot (Figure 2A). A total of 251 DEGs show a significant false discovery rate (FDR) (*q* < 0.05), and 140 of them had expressional changes of twofold or more in either direction. This list of differentially expressed transcripts was then used for further analysis. To obtain an overall assessment of the features of these DEGs, we used the online ShinyGO analytic tool [37]. First, we determined that the DEGs are primarily protein-coding RNAs (98.1%) and lincRNAs (1.9%), which is significantly different from the expected transcript distribution pattern (Figure 2B). This is consistent with the fact that EJC factors have little effect on small noncoding RNAs, such as miRNA and snRNAs. Second, DEGs from the P17 RNAseq dataset are generally evenly distributed across different chromosomes (Appendix A). However, we identified four regions in chromosomes 11 and Y that are enriched with DEGs (FDR < 0.05) (Appendix A). Interestingly, DEGs have longer coding sequences, transcript, 5′ untranslated region (UTR), 3′ UTR, and higher GC contents (Figure 2C–G). However, the number of exons (Appendix A) and the number of transcript isoforms per coding gene (Appendix A) were as expected in all genes.

To further examine which functions these differentially expressed transcripts mediate, we tested them in the Kyoto Encyclopedia of Genes and Genomes (KEGG) pathway [38] and looked for functional clusters that were enriched for genes in our DEG dataset. Consistent with our previous findings that *Rbm8a* is critical for interneuron development [31], the KEGG pathway analysis revealed that three major signaling pathways are enriched: neuroactive ligand–receptor interaction, complement and coagulation cascades, and the GABAergic synapse (Figure 3). DEGs relevant to neural functions are shown in the neuroactive ligand–receptor interaction (Appendix A) and the GABAergic synapse (Appendix A). Particularly, GABA-A receptor subunits, such as *Gabrd* and *Gabrq* genes, are enriched in the GABAergic synapse pathway, suggesting an imbalance of excitation and inhibition (E/I) that are prevalent in patients with neurodevelopmental disorders.

### 2.3. Gene Ontology (GO) Analyses on Upregulated and Downregulated DEGs at P17

This characterization of DEGs helps determine potential functions that can lead to changes in neurodevelopment by RBM8A. At P17, upregulated and downregulated CNS-related DEGs were examined with the GO Enrichment Analysis tool in ShinyGO. The downregulated DEGs are significantly enriched in multiple biological processes (Figure 4A,B), including fear response, chemical synaptic transmission, neuron development, neurogenesis, and transcriptional regulation. Using a network plot in which two pathways (nodes) are connected if they share 20% (default) or more genes, we detected two major clusters (Figure 4B). One regulates behavior and the other regulates neurodevelopment, which is consistent with the neurodevelopmental phenotypes of *Nes-Cre*; *Rbm8a^f/+^* mice (Figure 1). When DEGs were examined in the GO cellular component analysis, which is defined as “A location, relative to cellular compartments and structures, occupied by a macromolecular machine when it carries out a molecular function” in GO term [39,40,41], they are significantly enriched in several cellular compartments, such as dendrites, vesicle lumens, and neuronal spines (Figure 4C,D), suggesting a critical role of RBM8A in synaptic transmission.

Although the cellular distributions of DEGs are enriched in dendritic compartments, the GO molecular functions of these DEGs are clustered in transcriptional factors (Figure 4E,F). Among the downregulated group, two transcription factors stood out: *Neuronal differentiation 1* (*Neurod1*) and *Engrailed 2* (*En2*). *Neurod1* is a transcription factor critical for neurodevelopment [42,43]. It promotes neuronal cell phenotypes when overexpressed in stem cells, and in neurons [44,45,46]. *En2* also promotes the differentiation of neuronal subtypes [47,48,49]. With these observations, it is possible that *Rbm8a* is required for the activity of neural transcription factors, which allows more NPCs to remain in the progenitor pool and proliferate. Consistent with previous observations [29,31,32], if neuronal differentiation is impeded by *Rbm8a*, the competing process of brain development would be impaired.

Next, we investigated the upregulated DEGs at P17 in the GO analysis. Intriguingly, *Rbm8a* cKO significantly increases genes that participate in kidney development, blood vessel development, and ion transport (Figure 5A,B). Network analysis revealed two separate biological processes that are involved in ion transport and tube morphogenesis (Figure 5B). These results suggest that *Rbm8a* cKO in the nervous system suppresses the expression of neural genes, yet promotes other organ development, such as the renal system. Major cellular components were identified in the plasma membrane and extracellular matrix (Figure 5C,D). Interestingly, the serotonergic synapse, platelet alpha granule, and cell surface compartment are separated from other clusters in network analysis (Figure 5D). Consistent with GO cellular component analysis, upregulated DEGs are involved in active transmembrane transporters and growth factor binding (Figure 5E,F). Among the top upregulated DEGs, *transthyretin (Ttr)* encodes a homo-tetrameric carrier protein to transport thyroid hormones or vitamin A in the plasma and cerebrospinal fluid [50]. Mutations in Ttr can lead to several deadly diseases such as cardiomyopathy and neuropathy, which affect autonomic, motor, and sensory systems [51]. *Folate receptor 1* (*Folr1*) is a cell surface marker of midbrain dopaminergic neuron precursor cells and immature neurons of the same type [52]. These results further support the crucial role of RBM8A in neural and other organ development.

### 2.4. Alternative Splicing (AS) Analyses of RNAseq Dataset at P17

RBM8A is primarily known for its role in RNA regulation, including NMD and splicing. Therefore, we decided to investigate whether *Rbm8a* cKO led to changes in alternative splicing. We used MISO to determine if any alternatively spliced transcripts are significantly changed in our RNAseq results [53]. A total of 71 alternative splicing events in 62 genes were identified, with the majority being skipped exons (Figure 6A). Interestingly, the gene list did not overlap with any DEGs, suggesting that the levels of DEGs are not regulated by AS. The genes that were alternatively spliced were identified and input into GO analysis to determine if they mediate any biological functions. Intriguingly, the alternatively spliced genes in *Nes-Cre; Rbm8a^f/+^* mice at P17 affected functional pathways mediating gliogenesis, oligodendricyte development, and translational readthrough (Figure 6B,C). Together, these analyses reveal that RBM8A could regulate multiple neural functions and processes via controlling transcript abundance and AS.

### 2.5. DEG Analysis in the E12 Hindbrain

Our previous study conducted RNAseq analysis on the E12 cortex of control and *Nes-Cre; Rbm8a^f/+^* mice [31]. As the *Rbm8a* cKO mouse also has a small hindbrain (Figure 1B), we further tested the gene expression in the E12 hindbrains using RNAseq (Figure 7, supplemental Appendix A). We were interested in whether different groups of genes would be affected by *Rbm8a* cKO in the different brain regions. A volcano plot was generated to display all genes that had quantifiable readings in both the WT and KO hindbrains (Figure 7A). About 28,000 genes were plotted in the graph. A total of 25 DEGs had significant *q*-values (<0.05), and 23 of them had expressional changes of twofold or more in either direction (Figure 7A). The heatmap for these 23 DEGs was compared between the WT and KO mice in Appendix A. Similarly, these DEGs from E12 hindbrains are enriched in protein-coding genes (Figure 7B). Because the number of DEGs is low, they are localized in chromosomes 1, 2, 4, 5, 6, 7, 10, 11,17, X, and Y (Appendix A). Four enriched regions were identified in chromosomes 2, 6, and Y (Appendix A). The only significant feature of DEGs from E12 hindbrains is the 5′ UTR length compared to the overall genome (Figure 7C). No significant changes were identified in the number of exons (Appendix A), or the number of isoforms per coding gene (Appendix A). In contrast to the P17 whole brain data, DEGs from E12 hindbrains have normal lengths in the coding sequence (Appendix A), transcript length (Appendix A), 3′UTR (Appendix A), and normal GC content (Appendix A).

To further examine the functions of these DEGs, we tested them in the KEGG pathway [38]. Intriguingly, the KEGG pathway analysis revealed only one enriched major signaling pathway—the P53 pathway (Figure 7D, Appendix A)—suggesting a defect in the P53-mediated cell death pathway during embryonic neurodevelopment.

To examine the affected pathways, we further examined the DEGs from E12 hindbrain data in GO analysis. Among 25 DEGs, 8 DEGs are downregulated and no significant Biological Process is detected. We only identified some cellular components, such as translational initiation factor 2 complex and P granule (Figure 8A), and molecular function on histone H3 trimethylation (Figure 8C), in GO analyses. However, we were able to identify more enriched functions from upregulated DEGs (Figure 8C–H). Consistently, GO biological function analysis identified apoptosis, DNA damage, P53-mediated signal transduction, and epithelial cell maturation (Figure 8C,D), suggesting an increase in cell death during embryonic hindbrain. These DEGs are localized in various compartments (Figure 8E), but mainly in the two clusters centered in neuronal projection and protein kinase signaling complexes such as the TOR complex (Figure 8F). In addition to the kinase signaling pathways, GO molecular function analysis found more neural-related functions in dopamine β-mono-oxygenase activity, and opioid peptide activity (Figure 8G). These molecular functions are loosely connected in the network analysis (Figure 8H).

Compared to the E12 time point, even with hindbrain and cortex DEGs combined, many more genes showed significant expressional changes at P17. However, fewer genes overlapped between the P17 whole brain and the E12 brain regions than between the two E12 regions (Figure 9). *Nrgn* and *Anoctamin 3* (*Ano3*) were upregulated in the E12 cortex but in the opposite direction at P17. *Ano3* is a calcium-dependent phospholipid scramblase highly expressed in the brain and skin [54]. Meanwhile, *Top2a* was downregulated at both E12 and P17, whereas *Spp1* and *Gpnmb* were upregulated at both E12 and P17. These findings suggest that some downstream effects of *Rbm8a* cKO are temporally distinct, while others may underlie a long period of development in the CNS.

In all the time points/brain regions, *Fam212b* was significantly changed. However, the exact pathways implicating *Fam212b* are not yet known. In the embryonic brain, *Fam212b* is expressed by rapidly proliferating NPCs, while in the postnatal brain, it is expressed in limited, immature neuronal subtypes [55]. This increase in *Fam212b* could indicate a larger population of proliferating NPCs, contradicting our other findings, but it could also be the product of a compensatory mechanism among a dwindling pool of NPCs.

Overall, when we compared the hindbrain dataset with our E12 cortex dataset, fewer DEGs were significant at any level in the E12 hindbrain than in the cortex. Ten DEGs overlapped between those detected in the cortex and hindbrain; all of these were upregulated. Their names and functions are presented in Appendix A. Of note, six of these ten common upregulated DEGs are known to directly influence cellular proliferation. These were *Cdkn1a* [56], *Ccng1* [57], and *Phlda3* [58], which are known to slow or arrest the cell cycle. *Sesn2*, which protects cells from programmed death during stress [59]; *Eda2r*, which increases programmed cell death [60]; and *Fam212b* [55], which is highly expressed in rapidly proliferating NPCs in the embryonic mouse brain.

## 3. Discussion

In this study, three RNAseq datasets were analyzed to explore the altered transcriptome of *Rbm8a* cKO mice. Transcriptomes were assessed at E12 and P17, and at E12, the brain was split into cortex and hindbrain for separate sequencing. The results showed that the different brain regions and time points had many expressional changes, with little overlap between them. Therefore, loss of *Rbm8a* has temporally and spatially restricted effects during CNS development.

At E12, in the cortex, 19 DEGs significant at *q* < 0.05 were known to be implicated in the CNS [31]. They affect many aspects of nervous system development ranging from cell proliferation to myelin maintenance to calcium signaling. The hindbrain at E12 shared ten upregulated DEGs with the cortex, more than half of which could modulate the rate of cell proliferation and turnover. Some of them were pro-apoptotic and some were anti-apoptotic, while others regulated the progression of the cell cycle. Based on this data alone, it is not possible to conclude whether cell populations increased or decreased. However, the small body size and microcephaly of the mice suggest that the cells were less proliferative or more prone to dying [31].

At P17, a much different set of CNS-related DEGs was identified. Significant *Neurod1* and *En2* upregulation at P17, as well as downregulation of several genes associated with the immature CNS, indicates that neurons were possibly reaching terminal differentiation long before the CNS should have stopped developing. There was also evidence that the distribution of cell types was abnormal in the *Rbm8a* cKO brains, based on the decrease in *Lhx8* expression, which regulates the NPC’s decision to differentiate into a GABAergic versus a cholinergic neuron [61,62,63]. These results correlate with our previous findings that *Rbm8a* generally suppresses NPC differentiation. Apparently, loss of *Rbm8a* may also disrupt the ratios of NPCs that differentiate into each neuronal subtype.

A few of the significant DEGs from E12 reappeared in the P17 cKO brains. Notably, three of them had changed significantly at both E12 and P17. *Spp1* and *Gpnmb* were upregulated at both ages in cKO than control mice, while *Fam212b* was downregulated at P17 and upregulated at E12. This supports that some pathways are not continuously active, but rather are active on different timelines. Interestingly, both *Spp1* and *Gpnmb* play important roles in microglia and macrophage during brain damage and many other pathological conditions [64,65,66,67]. Upregulation of *Spp1* and *Gpnmb* indicates activation of microglia and neuroinflammatory responses in *Rbm8a*-deficient brains [68]. Their expressional changes could also be compensatory for other disruptions in the CNS. Additionally, both *Spp1* and *Gpnmb* participate in bone and tissue remodeling [69].

*Fam212b* was the only DEG that is upregulated at E12 but downregulated at P17 (*q* < 0.05). According to previous explorative studies, *Fam212b* is expressed by highly proliferative NPCs, immature neurons in the postnatal developing brain, and very specific subtypes of mature neurons in the adult forebrain [55]. Unfortunately, the exact pathways that this protein participates in are unknown. Further investigation is necessary to elucidate the role of *Fam212b* in CNS development, and its relation to *Rbm8a*.

Enrichment analysis showed that several pathways were affected by *Rbm8a* cKO in the brain. A few patterns that appeared across the three RNAseq datasets were enrichments in genes related to cellular differentiation, regulation of RNA transcription, proliferation, and cell death. Changes in differentiation pathways can result in delayed differentiation, premature differentiation, or an unbalanced distribution of cell types at maturity. Among enriched and depleted pathways, cell fates including oligodendrocytes, osteoblasts, neurons, and specific neuronal subtypes were named. Considering that several genes expected to be expressed in the adult brain were upregulated in the embryonic cortex, as well as the fact that negative regulation of photoreceptor differentiation was depleted, we hypothesized that the *Rbm8a* cKO mouse nervous system differentiates prematurely, resulting in the underdevelopment of nervous system tissues.

Closely tied to differentiation is the renewal of progenitor cell populations, regulated by signals for cell cycle progression versus arrest, and survival versus apoptosis. In the E12 cortex, genes for the cell division process were depleted; likewise in the hindbrain, negative regulation of proliferation was increased, and neural precursor proliferation was specifically determined to be depleted. This falls in line with our previous observations that *Rbm8a* promotes the renewal of NPCs and inhibits the differentiation of neuronal subtypes.

In the P17 brain, it appears that the nervous system gets a head start and develops quickly in *Rbm8a* cKO mice: neuronal development genes are enriched, and pathways pertaining to synaptic plasticity and behavior are more active. However, these could also be the results of premature differentiation of neurons. At a stage when the nervous system should still be expanding, the neurons are settling into their mature roles, approaching terminal differentiation. Furthermore, synaptic plasticity and behavior changes are observed in both juvenile and adult animals. Increased activity of these pathways is not necessarily an advantage for animals at such an early developmental stage. Intriguingly, *Rbm8a* cKO mice die at the postnatal stage even with another intact copy of the *Rbm8a* gene, which is different from human patients with 1q21 deletion or TAR syndrome who can live to adulthood. Although the mouse model can recapitulate some aspects of human disease, species variances between human and mouse models exist. This difference could be a lack of unknown compensatory mechanisms in mice.

RBM8A modulates mostly protein-coding genes that likely play a large role in the observed phenotypes, but RBM8A also regulates a proportion of lincRNAs. In the future, the location of the lincRNAs should be further investigated to determine which protein-coding genes they potentially modulate. This insight may lead to clues to the overall mechanism of RBM8A’s developmental role.

Taken together, the DEG analysis and GO enrichment analysis support our hypothesis that RBM8A maintains renewal of the neural precursor population and inhibits differentiation. Additionally, we uncovered specific genes and pathways for further investigation that may be critical to early CNS development. Finally, our RNAseq analysis featured several genes whose functions have not been elucidated in the context of early brain development, including *Spp1*, *Gpnmb*, and *Fam212b*. We hope that these data will provide the lead for further studies of brain development in mice and other mammalian models.

## 4. Materials and Methods

### 4.1. Mice

Wild-type male and female *C57/BL6N* mice were obtained from Taconic (Germantown, NY, USA) *C57BL/6N* male mice were housed 2–4 mice per cage in a room with a light/dark cycle at 12 h intervals (lights on at 7:00 am), and provided ad libitum access to food and water. All procedures on mice were reviewed and approved by The Pennsylvania State University IACUC Committee, under IACUC protocol, 44057, to Yingwei Mao.

### 4.2. RNA-Sequencing

Sample preparation for RNA sequencing was done by Dr. Yingwei Mao. Eight mouse embryos at E12 were collected for RNA sequencing. Four of them were *Rbm8a^fl/+^*, and the other four were *Nes-Cre*; *Rbm8a^fl/+^*. The hindbrain and cortex regions were dissected from the rest of the brain and stored separately. Six more mice, three for each condition, were euthanized on postnatal day 17 (P17); their whole brains were collected. These three sets of brain samples were sent to the Penn State Genomics Core Facility for sequencing with the Illumina HiSeq 2500 on a paired-read protocol. A total of 20 million paired reads were run per sample, producing 40 million total reads per sample. Raw reads were processed with paired-end analysis.

### 4.3. Analysis of DEGs

Three sequencing datasets were obtained, corresponding to the E12 cortex, E12 hindbrain, and P17 whole brain. The raw Illumina output was processed by the Penn State Bioinformatics Consulting Center, in collaboration with Dr. István Albert. Using TopHat (version 2.0.6), reads were aligned to the NCBI *Mus musculus* genome, assembly GRCm38.p6, available to the public through the NCBI Genome database. Subsequently, Cuffdiff was used to calculate the statistical significance of expressional changes.

After sorting DEGs by significance, DEGs were compared between the E12 cortex and hindbrain regions, as well as between the E12 and P17 time points. We identified genes that were significant at *q* < 0.05 in both conditions being compared and noted whether these shared DEGs had changed in the same direction.

The E12 cortex and P17 DEGs were further sorted to distinguish those pertinent to the CNS and establish targets of interest for further investigation in *Rbm8a* cKO animals. The CNS-related DEGs of the E12 cortex were categorized manually, based on literature reports of their known functions and expressional patterns. This was less feasible for the large number of DEGs at P17 because the analysis named all CNS-related genes it recognized from the submitted DEGs. Therefore, we instead used the Gene Ontology (GO) enrichment analysis tool to classify CNS-related genes DEGs in the P17 data.

### 4.4. Analysis of Enriched Gene Clusters

Overrepresented gene clusters and pathways were identified among significantly upregulated and downregulated DEGs using the Gene Ontology Consortium’s free online resource, GO enrichment analysis [39,40], and the ShinyGO analytic tool [37]. GO enrichment analysis groups genes by function and pathway, then estimates how many genes from each group are expected in a list of a given number of genes. If the actual number of genes from the same group greatly exceeds the expected number, then that group of genes is determined to be enriched. The software requires an input list with a sufficient number of genes to accurately identify gene cluster enrichments; we began by inputting the DEGs significant at *q* < 0.05. The E12 cortex and hindbrain and the P17 whole brain were analyzed individually, with inputted DEGs further separated by direction of change (upregulation or downregulation). The PANTHER Overrepresentation Test was used to recognize groups of genes within the DEGs that occurred at significantly higher or lower counts than expected, relative to all known expressional patterns in the mouse genome.

### 4.5. Alternative Splicing Analysis

For the alternative splicing analysis, all bam files created by TopHat [70] were merged into a single file using samtools (version 1.1) [71]. The total number of reads that support the individual variants associated with each of the predicted functional alternative splicing events was determined using the MISO (Mixture of Isoform) package (version 0.5.3) [53] using events annotated as of 26 June 2013. Significant differentially spliced events were determined by requiring a Bayes’ factor > 10 and Δψ > 0.2 in a comparison of control and *Rbm8a* cKO. Each event was required to pass the default MISO minimum read coverage thresholds.

## Figures and Tables

**Figure 1 ijms-24-04600-f001:**
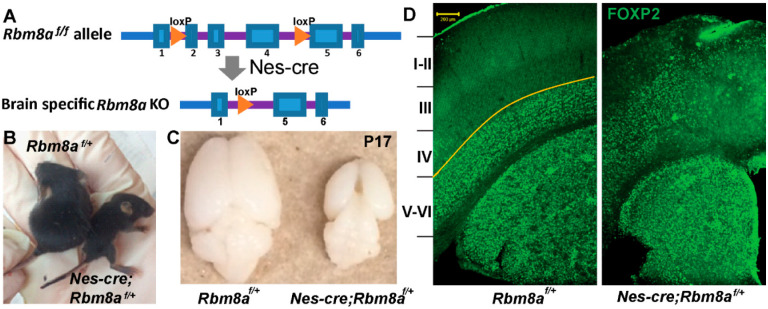
*Rbm8a* cKO design and mouse phenotypes. (**A**) The diagram presents a simple overview of the disruption of the *Rbm8a* allele. Cre recombinase excises exons 2, 3, and 4, disabling the gene. (**B**) *Nes-Cre; Rbm8a^f/+^* mice are smaller than littermate controls at P17 in body size. (**C**) *Nes-Cre; RBM8a^f/+^* mice suffer from microcephaly (small brain). (**D**) Coronal sections of P17 *Nes-Cre; Rbm8a^f/+^*, and *Rbm8a^f/+^* mice brains were immunostained for the deep cortical layer marker FOXP2 (green). Scale bar = 200 µm.

**Figure 2 ijms-24-04600-f002:**
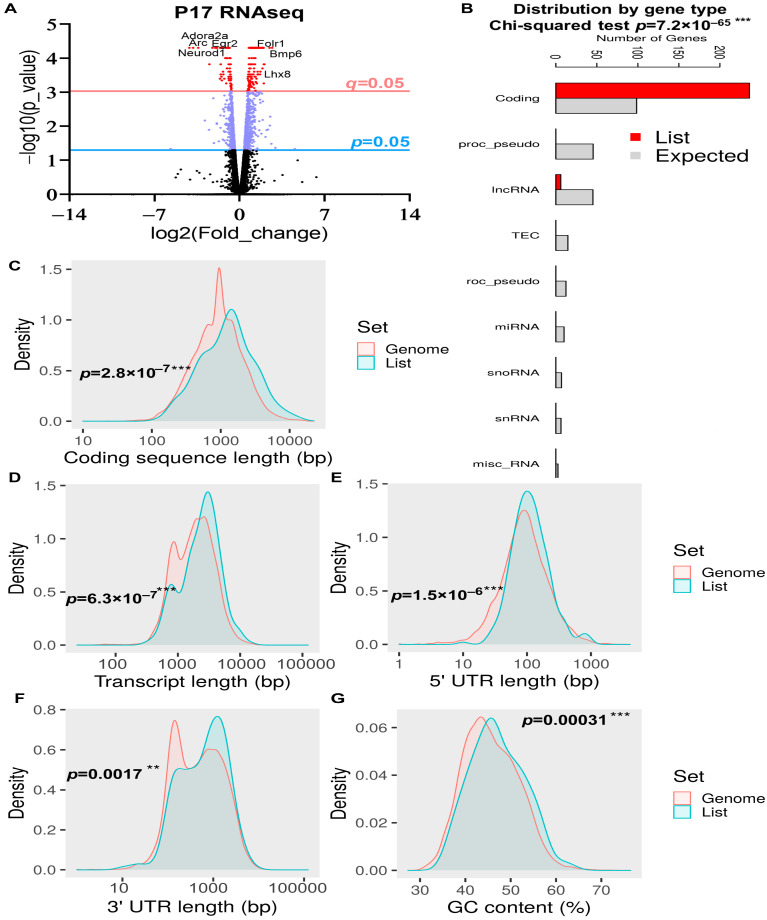
Quantifiable expressional changes in a whole-brain comparison between control and *Rbm8a* cKO mice at P17. (**A**) The volcano plot of all transcripts detected in the brains of control and *Rbm8a* cKO mice at P17. A total of 1227 genes were expressed differently with *p* < 0.05, 597 genes were expressed differently with *p* < 0.01, and 251 genes were expressed differently with *q* < 0.05. A total of 140 DEGs with significant *q*-values were up- or downregulated at least twofold. The *p* and *q* cutoffs = 0.05 are shown. (**B**) The distribution of DEGs is significantly enriched in the protein-coding genes. (**C**–**G**) DEGs at P17 show significant changes in coding sequence length (**C**), transcript length (**D**), 5′ UTR length (**E**), 3′ UTR length (**F**), and GC content (**G**) when compared with the general genome. **, *p* < 0.01; ***, *p* < 0.001 for (**B**–**G**).

**Figure 3 ijms-24-04600-f003:**
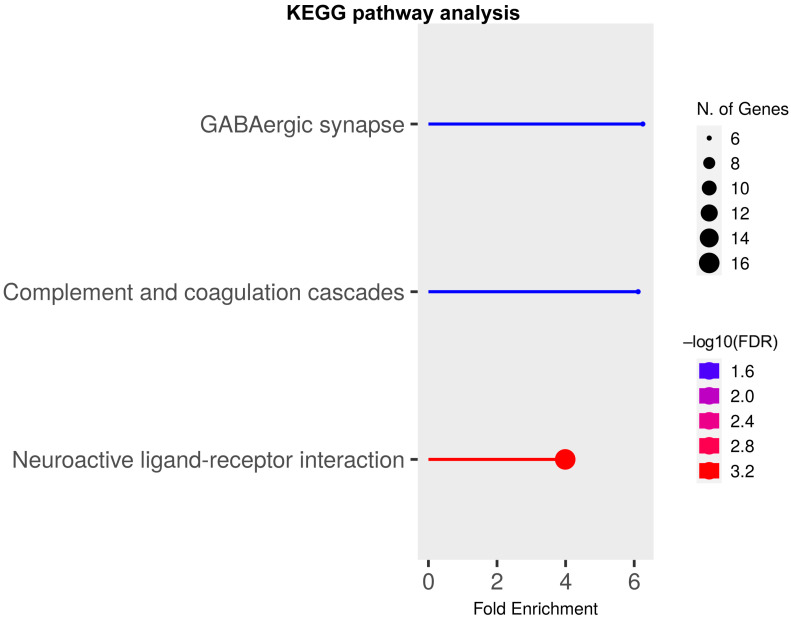
KEGG pathway analysis of DEGs at P17. Functional clusters are enriched for DEGs at P17. Color indicates the value of −log10 (FDR) and circle size indicates the enrichment of gene number.

**Figure 4 ijms-24-04600-f004:**
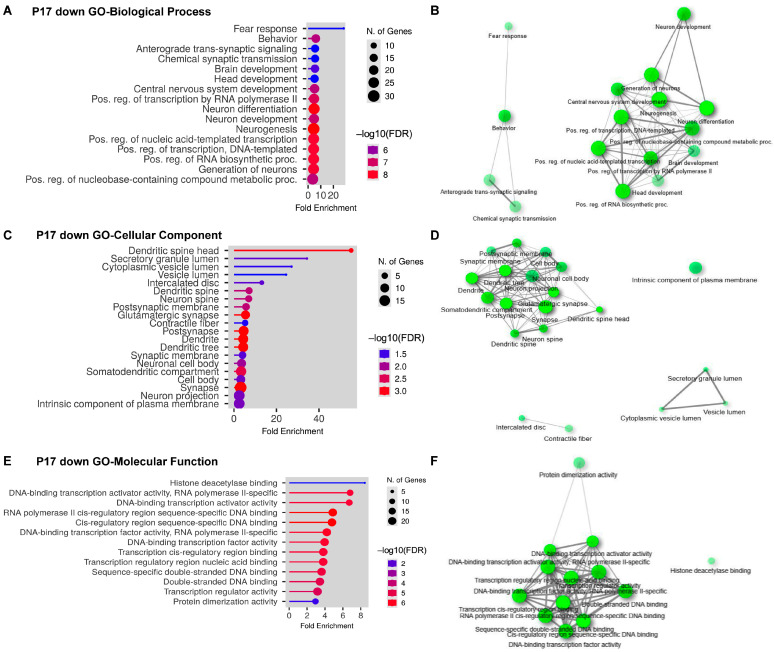
GO analysis of DEGs that are downregulated at P17. (**A**) Bar dot plot of GO analysis of biological process on downregulated DEGs from brains at P17. (**B**) Network plot of GO analysis of biological process on downregulated DEGs from brains at P17. (**C**) Bar dot plot of GO analysis of cellular component on downregulated DEGs from brains at P17. (**D**) Network plot of GO analysis of cellular component on downregulated DEGs from brains at P17. (**E**) Bar dot plot of GO analysis of molecular function on downregulated DEGs from brains at P17. (**F**) Network plot of GO analysis of molecular function on downregulated DEGs from brains at P17.

**Figure 5 ijms-24-04600-f005:**
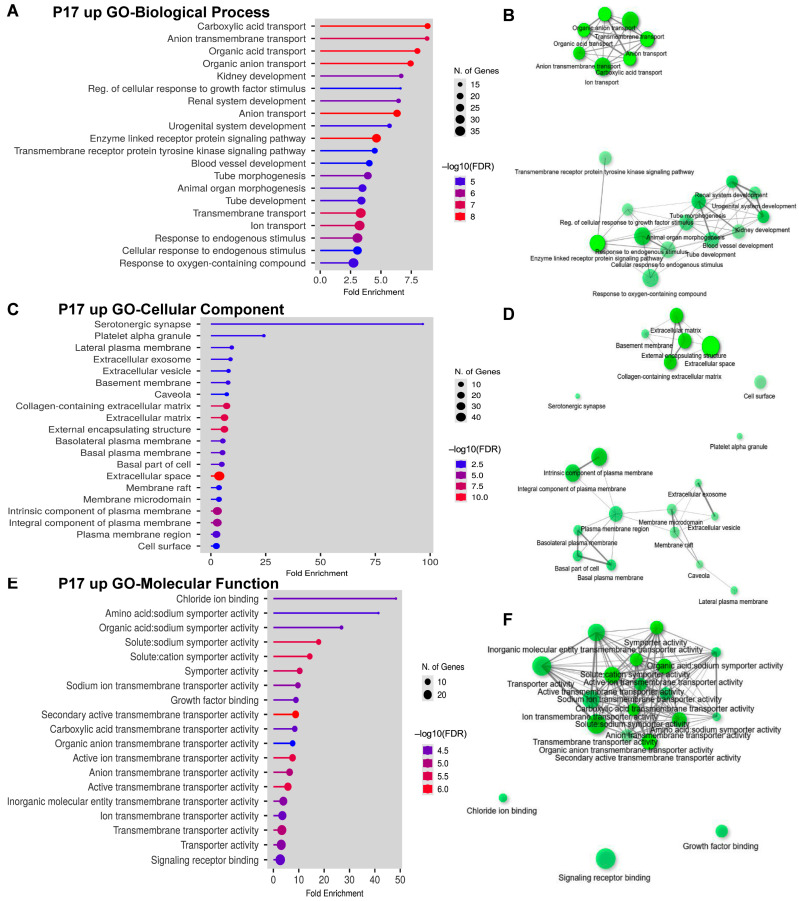
GO analysis of DEGs that are upregulated at P17. (**A**) Bar dot plot of GO analysis of biological process on upregulated DEGs from brains at P17. (**B**) Network plot of GO analysis of biological process on upregulated DEGs from brains at P17. (**C**) Bar dot plot of GO analysis of cellular component on upregulated DEGs from brains at P17. (**D**) Network plot of GO analysis of cellular component on upregulated DEGs from brains at P17. (**E**) Bar dot plot of GO analysis of molecular function on upregulated DEGs from brains at P17. (**F**) Network plot of GO analysis of molecular function on upregulated DEGs from brains at P17.

**Figure 6 ijms-24-04600-f006:**
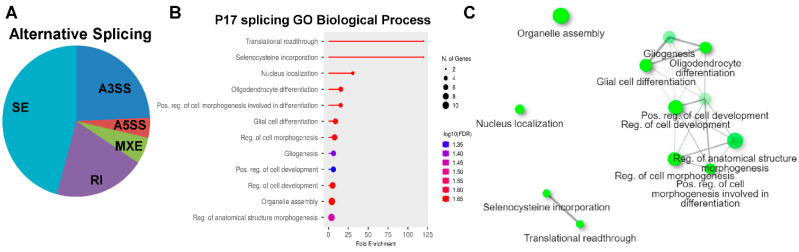
RBM8A is a moderate regulator of AS at P17. (**A**) MISO was used to AS events between the cKO and control datasets. RBM8A was found to be a moderate regulator of alternative splicing, as 62 mRNAs show significant changes in the AS in the brains of *Nes-Cre; Rbm8a^f/+^* mice at P17. A3SS: alternative 3′ splice site; A5SS: alternative 5′ splice site; MXE: mutually exclusive exons; RI: retained intron; SE: skipped exon. (**B**) GO analysis of AS genes at P17. (**C**) Network plot of GO analysis of biological process on AS genes at P17.

**Figure 7 ijms-24-04600-f007:**
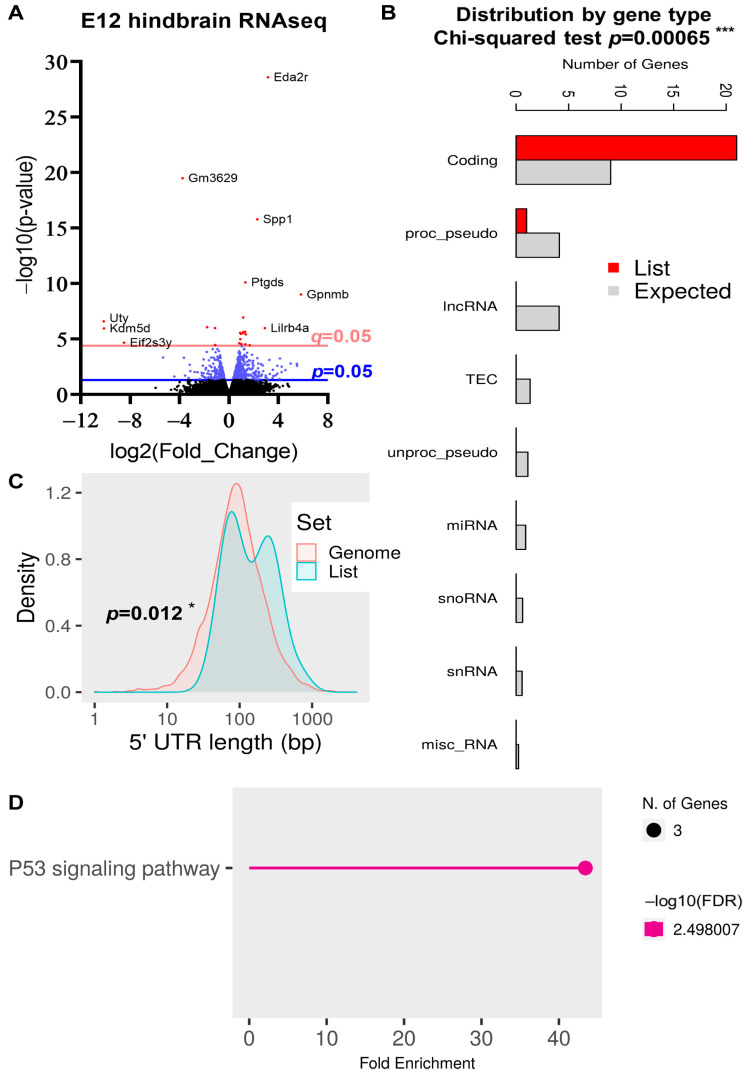
Quantifiable expressional changes in the hindbrain between control and *Rbm8a* cKO mice at E12. (**A**) The volcano plot of all transcripts detected in the brains of control and *Rbm8a* cKO mice at E12. A total of 847 genes were expressed differently with *p* < 0.05, 281 genes were expressed differently with *p* < 0.01, and 25 genes were expressed differently with *q* < 0.05. A total of 23 DEGs with significant *q*-values were up- or downregulated at least twofold. The *p* and *q* cutoffs (=0.05) are shown. (**B**) The distribution of DEGs is significantly enriched in protein-coding genes. ***, *p* < 0.001. (**C**) DEGs at the E12 hindbrain show significant changes in the length of 5′UTR. *, *p* < 0.05. (**D**) KEGG pathway analysis of DEGs of E12 hindbrains.

**Figure 8 ijms-24-04600-f008:**
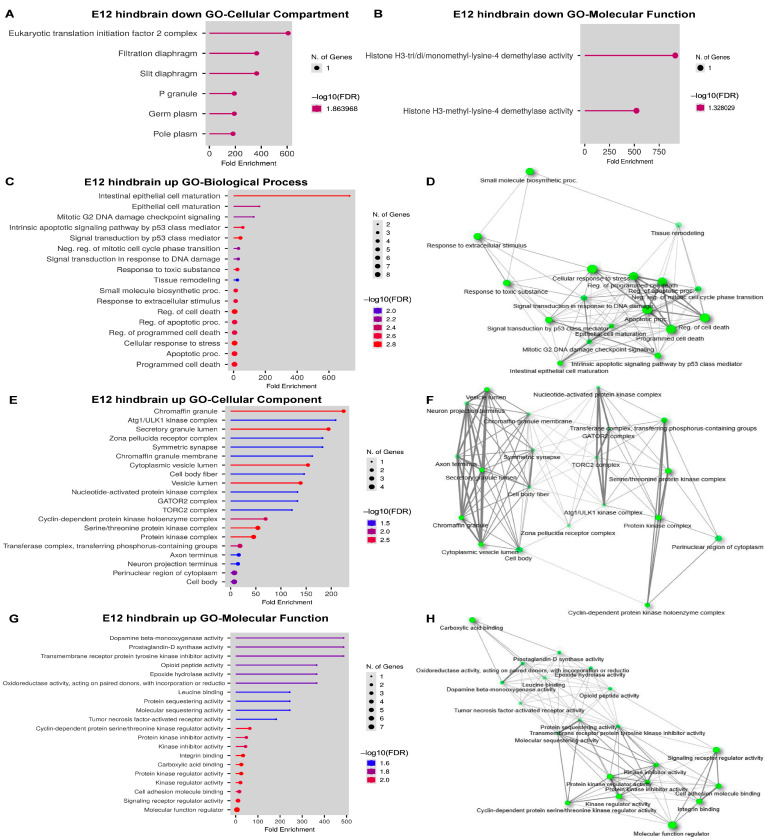
GO analysis of DEGs from E12 hindbrain. (**A**) Bar dot plot of GO analysis of cellular component on downregulated DEGs from E12 hindbrains. (**B**) Bar dot plot of GO analysis of molecular function on downregulated DEGs from E12 hindbrains. (**C**) Bar dot plot of GO analysis of biological process on upregulated DEGs from E12 hindbrains. (**D**) Network plot of GO analysis of biological process on upregulated DEGs from E12 hindbrains. (**E**) Bar dot plot of GO analysis of cellular component on upregulated DEGs from E12 hindbrains. (**F**) Network plot of GO analysis of cellular component on upregulated DEGs from E12 hindbrains. (**G**) Bar dot plot of GO analysis of molecular function on upregulated DEGs from E12 hindbrains. (**H**) Network plot of GO analysis of molecular function on upregulated DEGs from E12 hindbrains.

**Figure 9 ijms-24-04600-f009:**
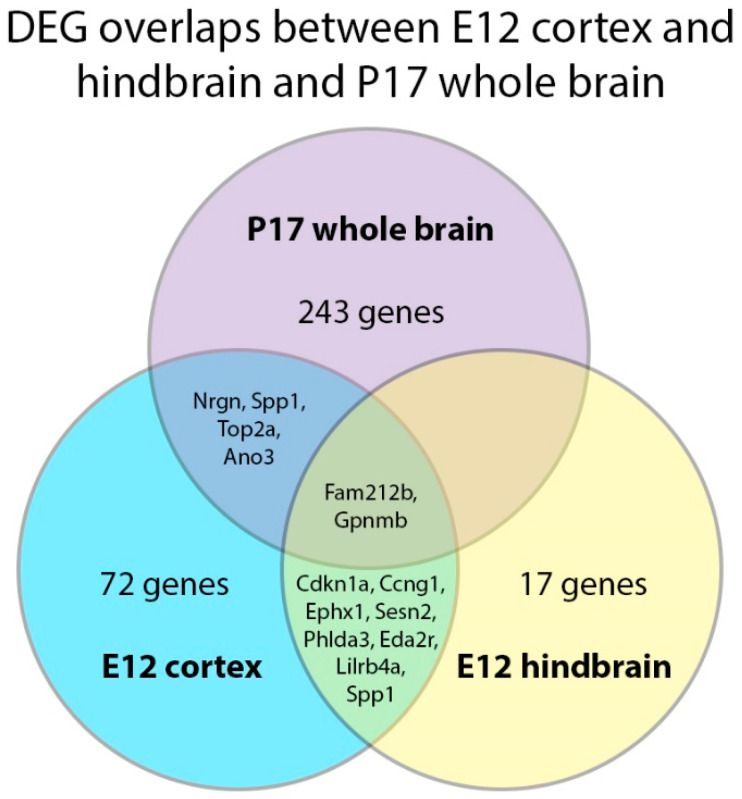
Overlaps between significant DEGs of the E12 cortex, E12 hindbrain, and P17 whole brain. DEGs with significant *q*-values were compared across the three RNAseq datasets. Very few DEGs overlapped between the P17 and E12 time points.

## Data Availability

All RNAseq raw data will be submitted to NCBI (Bioproject PRJNA631303) upon manuscript acceptance.

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
