# Peer review of "Transcriptomic Analyses of Brains of RBM8A Conditional Knockout Mice at Different Developmental Stages Reveal Conserved Signaling Pathways Contributing to Neurodevelopmental Diseases"

_ijms, 2023, doi:10.3390/ijms24054600_

Round 1
Reviewer 1 Report
1. All figures inserted in the manuscript are very blurry, and I cannot read them except figure 1.
2. Nestin was described as a marker of neuroepithelial stem cells. However, there are reports of Cre recombinase activity in a few isolated kidney and heart cells. So is Rbm8a also KO in these tissues, other than in the brain? Moreover, is Rbm8a expressed in neurons and/or glial cells?
3. The authors didn’t validate the KO mice, like the reduction in mRNA expression or protein expression level. Figure 1A can be improved, I even didn’t see STOP between two loxP sites. In Figure 1D, the authors need to draw out layers 5/6.
4. The authors mentioned that “Most of these Nes-cre; Rbm8af/+ pups only survived until postnatal day 20 (P20)” (Line 126-127). Though RBM8A mutations have been implicated in a variety of clinical phenotypes (Line 60), it did not cause death in humans. Therefore, in this manuscript, the brain-specific KO of Rbm8a did not mimic any situations in human, and I don’t think it would make any sense to RBM8A mutation-induced clinical diseases.
5. What is the full name of NPC? Line 81.
Author Response
We thank for reviewer’s insightful comments. The modifications are highlighted in yellow. We addressed the comments point-by-point below:
- All figures inserted in the manuscript are very blurry, and I cannot read them except figure 1.
--Thanks for the comments. We have We have increased resolution and contrast and uploaded high resolution figures separately in the revision to help reviewers.
- Nestin was described as a marker of neuroepithelial stem cells. However, there are reports of Cre recombinase activity in a few isolated kidney and heart cells. So is Rbm8aalso KO in these tissues, other than in the brain? Moreover, is Rbm8a expressed in neurons and/or glial cells?
--Thanks for the comment. Nestin-cre is expressed in a few cells in theheart and kidney but it will not affect our conclusion as our focus of this study is in the brain. Rbm8a is expressed in both astrocytes and neurons in our hand (data not shown) and other public available datasets, such as https://www.proteinatlas.org/ENSG00000265241-RBM8A/single+cell+type/brain. We have added more in discussion of this non-brain expression of Nestin cre line highlighted in yellow.
- The authors didn’t validate the KO mice, like the reduction in mRNA expression or protein expression level. Figure 1A can be improved, I even didn’t see STOP between two loxP sites. In Figure 1D, the authors need to draw out layers 5/6.
--Thanks for the comment. We have fully validated our KO model in our previous publication in Ref-32 (Translational Psychiatry 2020, 10, 379). In that paper, we have demonstrated the depletion of RBM8A protein in both culture cells and brain tissues. As the knockout of RBM8A is dependent on cre recombinase activity not the STOP cassette, there is no need to add the STOP. Please refer to the supplemental Figure S1 in our paper Ref-32 for details of this cKO line. This study is a nature extension of our previous paper to try to better understand the molecular mechanism. We have modified Figure 1 D to add the layers 5/6.
- The authors mentioned that “Most of these Nes-cre; Rbm8af/+ pups only survived until postnatal day 20 (P20)” (Line 126-127). Though RBM8A mutations have been implicated in a variety of clinical phenotypes (Line 60), it did not cause death in humans. Therefore, in this manuscript, the brain-specific KO of Rbm8a did not mimic any situations in human, and I don’t think it would make any sense to RBM8A mutation-induced clinical diseases.
--Thanks for the insightful comments. Ours and other group’s studies showed consistent results that mouse Rbm8a KO phenotypes are more severe than human probably due to lack of some unknown compensatory mechanism. Although all mouse KO models including ours may not be able to mimic every aspect of human diseases, mouse models still can provide molecular insights for better understanding human related diseases, not limited to neurological diseases. Moreover, our studies are pure basic sciences to examine molecular mechanisms related to Rbm8a functions. We have added more in the discussion on this point highlighted in yellow.
- What is the full name of NPC? Line 81.
--Thanks for the comment. We have spelled out the abbreviation of NPC highlighted in yellow.
Reviewer 2 Report
RBM8A mutations have been implicated in a variety of clinical phenotypes, especially of neurological origin. To investigate how the downstream molecular mechanism of Rbm8a causes abnormal development of the brain at different developmental periods,the authors analyzed the changes in the transcriptome of mice with Rbm8a haploinsufficiency in the brain during embryonic and postnatal stages. They identified over 300 transcripts that showed significant fold changes between WT and Rbm8a cKO mice (which the authors have previously created), including 34 genes with known functions in nervous system development. The authors hope that this dataset will help them and others in choosing a narrower subset of genes or cellular processes to examine in future studies. Interestingly, their RNA-seq analysis found some genes whose functions have not been elucidated in the context of early brain development, including Spp1, Gpnmb, and Fam212b.
The study here is thoroughly conducted and the results obtained support the conclusion that the authors are drawing. The quality of the work is also suitable for this journal. My main comments are about the clarity of figures. The figures are generally small and could be improved by increasing the resolution.
-
The images that show chromosomal locations and pathways, for eg: Fig 2C, 3B, 3C, 7C and 7F could be included in the supplementary materials to enhance the quality of neighboring figures which are more important to the manuscript.
-
Similarly, the network plots need more clarity and better resolution (for eg: 4B, 4D, 4F and others). Could the authors please ensure that the text in these network plots are properly visible and clear.
Lastly, it will be beneficial to the reader if the term ‘cellular compartment’ is explained in the manuscript. Readers from different fields interpret compartments in different ways. For eg: CIliary compartment, any-organellar compartment and so on. It will be helpful if the authors clarify what they mean by compartment in this context.
Author Response
We thank for reviewer’s insightful comments. The modifications are highlighted in yellow. We addressed the comments point-by-point below:
- The images that show chromosomal locations and pathways, for eg: Fig 2C, 3B, 3C, 7C and 7F could be included in the supplementary materials to enhance the quality of neighboring figures which are more important to the manuscript.
--Thanks for the comments. We have modified the figure accordingly.
2. Similarly, the network plots need more clarity and better resolution (for eg: 4B, 4D, 4F and others). Could the authors please ensure that the text in these network plots are properly visible and clear.
--Thanks! We have increased resolution and contrast and uploaded high resolution figures separately in the revision to help reviewers.
Lastly, it will be beneficial to the reader if the term ‘cellular compartment’ is explained in the manuscript. Readers from different fields interpret compartments in different ways. For eg: CIliary compartment, any-organellar compartment and so on. It will be helpful if the authors clarify what they mean by compartment in this context.
--Thanks! We have added more explanations on different terms from GO analyses including ‘cellular component’.
Reviewer 3 Report
Dr. Mao and colleagues reported a very interesting study elaborating that the loss of Rbm8a causes decreased cellular proliferation, increased apoptosis and early differentiation of neuronal subtypes, which may lead ultimately to an altered neuronal subtype composition. The manuscript was very well-presented. The study design was impressive, and results were well documented. Statistical and bioinformatics analysis were thorough and reliable. This study is clearly relevant to the field. I believe this article should be considered for publication.
Author Response
We thank the reviewer for the enthusiastic comments!
Round 2
Reviewer 1 Report
Agree with being accepted.